# Cognitive Outcomes and Relationships with Phenylalanine in Phenylketonuria: A Comparison between Italian and English Adult Samples

**DOI:** 10.3390/nu12103033

**Published:** 2020-10-03

**Authors:** Cristina Romani, Filippo Manti, Francesca Nardecchia, Federica Valentini, Nicoletta Fallarino, Claudia Carducci, Sabrina De Leo, Anita MacDonald, Liana Palermo, Vincenzo Leuzzi

**Affiliations:** 1School of Life and Health Sciences, Aston University, Birmingham B4 7ET, UK; 2Department of Human Neuroscience—Unit of Child Neurology and Psychiatry, Sapienza University of Rome, 00185 Rome, Italy; filippo.manti@uniroma1.it (F.M.); francesca.nardecchia@uniroma1.it (F.N.); vincenzo.leuzzi@uniroma1.it (V.L.); 3Department of Psychology, Sapienza University of Rome, 00185 Rome, Italy; valentini.fe@gmail.com (F.V.); fallarino.1707273@studenti.uniroma1.it (N.F.); 4Department of Experimental Medicine, Sapienza University of Rome, 00185 Rome, Italy; claudia.carducci@uniroma1.it; 5Department of Clinical Medicine, Policlinico Umberto I, 00161 Rome, Italy; sabrina.deleo@libero.it; 6Birmingham Women’s and Children’s NHS Trust, Birmingham B15 2TG, UK; Anita.Macdonald@nhs.net; 7Department of Medical and Surgical Sciences, Magna Graecia University of Catanzaro, 88100 Catanzaro, Italy; liana.palermo@unicz.it

**Keywords:** PKU, cognitive outcomes, cross-cultural, cross-countries, Phe associations

## Abstract

We aimed to assess if the same cognitive batteries can be used cross-nationally to monitor the effect of Phenylketonuria (PKU). We assessed whether a battery, previously used with English adults with PKU (AwPKU), was also sensitive to impairments in Italian AwPKU. From our original battery, we selected a number of tasks that comprehensively assessed visual attention, visuo-motor coordination, executive functions (particularly, reasoning, planning, and monitoring), sustained attention, and verbal and visual memory and learning. When verbal stimuli/or responses were involved, stimuli were closely matched between the two languages for psycholinguistic variables. We administered the tasks to 19 Italian AwPKU and 19 Italian matched controls and compared results from with 19 English AwPKU and 19 English matched controls selected from a previously tested cohort. Participant election was blind to cognitive performance and metabolic control, but participants were closely matched for age and education. The Italian AwPKU group had slightly worse metabolic control but showed levels of performance and patterns of impairment similar to the English AwPKU group. The Italian results also showed extensive correlations between adult cognitive measures and metabolic measures across the life span, both in terms of Phenylalanine (Phe) levels and Phe fluctuations, replicating previous results in English. These results suggest that batteries with the same and/or matched tasks can be used to assess cognitive outcomes across countries allowing results to be compared and accrued. Future studies should explore potential differences in metabolic control across countries to understand what variables make metabolic control easier to achieve.

## 1. Introduction

Phenylketonuria (PKU) is an inherited metabolic disease occurring in about 1/10,000 live births where an error in the gene coding for the enzyme phenylalanine hydroxylase (PAH) produces an inability to metabolize the amino acid phenylalanine (Phe) into tyrosine with serious consequences for brain health [1]. Tremendous advances have been made in our understanding and treatment of this disorder from the 1930s when it was first discovered by a Norwegian physician, Asbjørn Følling, who noticed high levels of phenylpyruvic acid in the urine of some patients with severe mental disability and established connections with high levels of Phe in the blood. Since the wide-spread introduction of new-born screening in the late seventies in most countries, infants with PKU follow a Phe-restricted diet which lowers blood Phe levels and eliminates mental disability. It is now recommended that a PKU diet is followed throughout life. Current European guidelines recommend Phe to be kept within the target range of 120–360 μmol/L till 12 years of age and within 120–600 μmol/L, above 12 years [2,3]. American guidelines recommend a target range of 120–360 μmol/L throughout life (American College of Medical Genetics and Genomics, ACMG) [4]. In classical PKU, without treatment, Phe could exceed >2000 μmol/L.

Maintaining a Phe-restricted diet allows people with PKU to lead normal lives. However, not all is well. The PKU diet is expensive, unpalatable, and unsociable. Thus, it is often self-relaxed in late childhood and abandoned during late adolescence [5,6,7,8,9]. Possibly because Phe levels remain suboptimal, on average, people with PKU do not reach their full cognitive potentials [7]. IQ is in the normal range, but lower than matched controls [10], and there are impairments in cognitive tasks, especially when speed of processing and higher cognitive functions are involved [11,12,13,14,15,16]. Moreover, this is the first generation of early treated adults with PKU (from now on AwPKU) to reach middle-age and we do not know the effects of prolonged high levels of Phe on aging brains.

Better management of PKU may be achieved with the wider use of existing pharmacological treatments [3,17,18] and the introduction of new ones [19,20,21], but it also depends on a better understanding of how the cognitive impairments experienced by this population relate to levels and variations in Phe levels across the life span. While Phe may be particularly toxic for developing brains, we need evidence of the safety of accumulating high Phe levels on aging brains. Finally, it is important to understand whether some individuals are less affected by high Phe since there is some evidence of individual variation [6,22,23,24]. For a few people with PKU, it may be less important to keep on a strict diet.

Establishing the efficacy of new treatments and the safety of existing ones relies on the comprehensive cognitive assessments of large samples of people with PKU and relative controls. Ideally, cognition should be tested across the board because the effects of Phe may vary for different cognitive functions. Moreover, it is important to use multiple measures to increase reliability since tasks do not tap cognitive function in a simple, univocal way. However, this is time-consuming and recruiting participants is challenging since PKU is a rare disease. These difficulties, compounded by resource limitations, mean that it is difficult for studies based on single clinical centres to achieve the desired breadth and depth of testing with enough power [15,16,25] making the ability to collate results across national and international centres particularly important. However, there is a lack of studies comparing results across national samples.

Similar negative effects of PKU on cognition have already been reported across countries (for example, deficits in executive functions have been reported in: Italy: Nardecchia et al. [26]; the Netherlands: Jahja et al. [27]; the UK: Palermo et al. [15]; the USA: Brumm et al. [25]; Christ, et al. [28]; Diamond et al. [2]; deficits of speed of processing have been reported in Australia: Moyle et al. [10]; Germany: Feldmann et al. [29]; the UK: Channon et al. [30]; Palermo et al. [15]; the USA: Janos et al. [31]). These results suggest that the effects of Phe on cognition are similar across countries, in spite of cultural and environmental differences in the approach to food and feeding. However, there is a lack of studies directly comparing results collected using the same tests and direct comparisons are important to give us confidence that results can be accrued.

The objectives of the present study are twofold: 1. we aim to replicate results previously obtained with a relatively large sample of English AwPKU (*N* = 37) [5,6,15,32] by administering a subset of tasks to a new sample of Italian patients; 2. we aim to demonstrate that the same tasks which are sensitive to blood Phe in English are also sensitive in Italian so that they also demonstrate impairments and relationships with blood Phe levels and Phe fluctuations throughout the lifespan. A comparison between English and Italian PKU samples is particularly relevant given differences in the approach to food and diet in the two countries. Note that there is no issue of validity and specificity in the cognitive assessment of PKU. We do not need to distinguish people with PKU from healthy people. Genetic tests reliably establish the presence/absence of PKU from birth and high-level of Phe are constant in people of PKU if the disease is untreated. What is important, instead, is test-sensitivity to variations in metabolic control so that cognitive outcomes can be properly monitored. This can be demonstrated by showing impairment compared to healthy controls and correlations with metabolic measures.

We assessed metabolic control and cognitive outcomes in 19 Italian early-treated AwPKU and 19 matched controls and compared performance with that of 19 English early-treated AwPKU and relative controls selected from our previously tested cohort [15]. All groups were matched for age and education. Comparisons between PKU groups were in terms of z scores which considered performance in terms of deviations from the relative control groups. In addition, we assessed the sensitivity of our cognitive battery in Italian by assessing correlations with current and historical blood Phe levels. The Italian and English testing batteries were matched as rigorously as possible. In most cases, our tests were exactly the same (the same materials and procedure); those with verbal stimuli were carefully matched for psycholinguistic variables such as word frequency and word length (e.g., Rey AVLT). Similar levels of impairments and similar correlations with Phe levels will demonstrate test reliability and sensitivity for different national PKU samples. It will also give us confidence that, when the same or matched tasks are used, results can be accrued, allowing more power for analyses.

## 2. Method

### 2.1. Participants

All PKU participants were adults diagnosed soon after birth (2–3 days in Italy and 5–7 days in the UK).

Nineteen Italian AwPKU were recruited from the Clinical Centre for Neurometabolic Diseases in the Department of Human Neuroscience, Child Neurology and Psychiatry Unit at the Sapienza University of Rome. Three participants were currently treated with sapropterin. Nineteen Italian control participants were recruited among students and friends of the researchers. They were matched to the Italian PKU participants for age and education. Among the Italian participants, 4 had a diagnostic blood Phe level > 600 µmol/L but <1200 µmol/L; 15 participants had Phe > 1200 µmol/L at birth.

To allow a direct comparison between an Italian and an English sample, 19 English AwPKU were selected from a larger sample of 37 AwPKU previously tested [15,16,32]. They were all tested at the Inherited Metabolic Disease Unit at the Queen Elizabeth Hospital in Birmingham. They all had Phe > 1200 µmol/L at birth. They were matched one-to-one with the Italian AwPKU for gender, age, and education. Matching was blind to cognitive performance and metabolic control as possible differences were assessed. Thirty English healthy controls were originally recruited through an advertising volunteering website. From this sample, we blindly selected 19 healthy controls matching the English PKU participants for age and education (in number of years).

Power calculations indicated that 20 participants were necessary in the clinical group and 20 participants in the control group for a one tail effect size of 0.8 (consistent with what we found in our previous studies) and =0.05, power (1-error probability) = 0.80. All AwPKU treated in the English and Italian centres were invited to participate and were accepted in the study on a first-come, first-served basis. Recruitment stopped when enough participants were tested. After the Italian PKU participants were contacted, one participant became unavailable and we were left with 19 participants, which still gave our study acceptable power (=0.78).

The English study received NHS ethical approval. The Italian study was approved by the local ethics committee. All participants provided informed consent to the study.

### 2.2. Ethical Approval

The study was conducted in accordance with the Declaration of Helsinki. All participants gave their informed consent for inclusion before they participated in the study. In the UK, the protocol was approved by the West Midlands NHS Ethics Committee (rec REC: 10/H1207/115). In Italy the protocol was approved by the Institutional Review Board of “Sapienza”—University of Rome (Project identification code 3629).

### 2.3. Metabolic Measures

For both the English and the Italian participants, metabolic measures were taken regularly since birth and extensive records were available. The number of measures did not differ between countries (see Table 1). Blood Phe monitoring was performed on dry blood spot collected after overnight fasting by High Performace Liquid Chromatography until 2010 and then via tandem mass spectroscopy. The laboratories of both centres have adhered and contributed to international quality control systems

We averaged metabolic control in three age bands: childhood: 0–10 years old, adolescence: 11–16 years old, and adulthood: 17 years to present. We have also averaged measures throughout the life-time and considered current Phe levels. For the Italian group, current Phe has been measured immediately before the testing session/s or in the preceding few days; for the UK group, current Phe has been measured immediately before the two testing sessions and averaged. We considered two types of measures: Phe level and Phe fluctuation/variation (we will use the term fluctuation from now). Phe level in each band was calculated by taking the median values for each year and then averaging the year values; Phe fluctuation was calculated by taking the SD for each year and then averaging year values in the band.

### 2.4. Cognitive Assessment

Cognitive assessments were carried out in a quiet room at the clinical centres in Birmingham and Rome. The testing session for the Italian participants lasted between 2 and 3 h. The English participants carried out more tests and were tested in two separate sessions of similar length. Testing was carried out by a psychologist or a neuropsychiatrist with neuropsychological training.

IQ was measured, for the Italian participants, using the Wechsler Adult Intelligence Scale-Revised (WAIS-R) [33] and, for the English participants, the Wechsler abbreviated scale of intelligence (WASI) [34], which includes the following subtests: Vocabulary, Block Design, Similarities, and Matrix Reasoning. The WAIS-R and the WASI are strongly correlated providing similar validated measures of Full Scale IQ [35]. In addition, participants were given a set of tasks chosen from the larger set of tasks administered in our previous studies [15,16]. We chose tests which either showed a strong difference between PKU participants and controls and/or strong correlations with metabolic measures. We also prioritized tasks with non-linguistic stimuli which did not need adapting across languages. Therefore, we did not include tests of picture naming, reading, spelling, and orthographic knowledge (spoonerisms, phoneme deletions) where speed impairments could be due to a general reduction in speed of processing which was also tapped by visual search. In addition, we did not include tasks where relationship with metabolic measures are modest or absent [5]. Finally, to reduce the number of tasks tapping similar functions, we also did not administer the Tower of Hanoi, lexical learning task, the Stroop, and nonword repetition. Measures of short-term memory (digit span and Corsi span) and a baseline measure of peripheral speed of processing were included for completeness and to confirm or disprove impairments, given mixed results from the literature (for impairments in digit span and nonword repetition see Palermo et al. [15]; for contrasting results see Brumm et al. [25], and Moyle et al. [10]; see also Jahja et al. [27] for deficits with increasing working memory load).

The following cognitive areas were assessed:

#### 2.4.1. Visual Attention

This was assessed with four tasks [15,16]: 1. Simple Detection: Press a response button as soon as a ladybird appears on the screen; 2. Detection with Distractors: Press a button whenever a ladybird appears on the screen alone or with a green bug, in the second part of the task the instruction was changed to press a button whenever a green bug appeared on the screen alone or with a ladybird; 3. Feature Search: Detect a target among distractors not sharing features by pressing a “yes” or “no” button (e.g., a red ladybird among green bugs); 4. Conjunction Search: Detect a target among distractors sharing features (e.g., red ladybird among red bugs and green bugs). Both reaction times (RT from now on) and accuracy measures (error rates) were taken.

#### 2.4.2. Visuo-Motor Coordination

This was assessed with two tasks: 1. Grooved Pegboard Test [36]: Put pegs into the holes of a board using only one hand as quickly as possible (short version with two trials one with the dominant and one with the non-dominant hand for the Italian and English matched samples) and 2. Digit Symbol Task [33]: Fill as many boxes as possible with symbols corresponding to numbers (key with associations remains visible) in 90 s. Trail Making Test A (TMT A) [37,38]: connect circles containing numbers in ascending order of the numbers as quickly as possible.

#### 2.4.3. Complex Executive Functions

This was assessed with four tasks tapping skills such as planning, flexibility, and abstract thinking: 1. The Wisconsin Card Sorting Test (WCST) 64 card version [39]: Discover the rules to match cards from a deck with four reference cards according to the shape, number or colour of the symbols on the card; feedback is provided to allow learning. Flexibility is required when the sorting rule is changed unknown to the participant and the new rule must be discovered. We used three different scores: total errors, number of perseverative responses and number of completed categories. 2. Difference in speed between Trail Making Test B-A (TMT B-A) [37,38]. A involves connecting circles containing numbers in ascending order; B also involves connecting circles in ascending order but alternating between the number and letters contained in the circles. Only time is considered in this test; when occasionally an error is made, it is corrected by the examiner and this affects the time to complete the task. 3 Fluency: For letter fluency: generate as many words as possible starting with a given letter in a minute of time (for Italian: P, F and L; Novelli et al. [40] for English: C, F and L; Benton et al. [41]); for semantic fluency [42,43]: generate as many names of animals as possible in 1 min of time. This requires planning an efficient search through the lexicon.

#### 2.4.4. Short-term Memory/Working Memory

This was assessed with two tasks: 1. Digit Span: Repeat a sequence of digits spoken by the examiner soon after presentation; 2. Corsi Block Tapping Test [44]: The examiner taps a sequence of blocks and the participant must reproduce the sequence in the same order. Span was calculated as the longest sequence which could be repeated correctly (1 point was awarded for each length if all trials are correct; otherwise a corresponding fraction of point was awarded; for digit span where sequences started from length four, sequences of length 1–3 were assumed all correct, unless there were errors with sequences of four digits; in this case, sequences of shorter length were presented).

#### 2.4.5. Sustained Attention

This was assessed with the Rapid Visual Information Processing task (RVP) [45]: detect three target sequences of three digits by pressing the response key when the last number of the sequence appears on the screen. Scores are percentage correct.

#### 2.4.6. Verbal Memory and Learning

This was assessed with The Rey Auditory Verbal Learning Test [46,47] which asks for learning, immediate recall, and delayed recall of a list of 15 words. The list is presented five times and participants are asked to recall the words immediately after each presentation. After the 5th presentation (A5), an interfering list (B1) is presented and participants are asked to recall this list and then, once again, the original list (A6) without a further presentation. Finally, participants are asked to recall the original list after a 20-min filled interval. Our scores include total number of errors across the five learning trials (A1-5); errors in recalling the words after an interfering list (A6); and, again, errors in delayed recall of the original list.

#### 2.4.7. Visual Memory and Learning

This was assessed with the Paired Associates Visual Learning [48]: Learn to associate objects with locations. Z scores for each participant for each task were computed using the relevant control group as a reference point. As well as considering performance in individual tasks, for each participant, we computed two indexes of overall cognitive performance: 1. We averaged z scores in all tasks and 2. We considered the rate of poor scores across all tasks (rate of Z scores => 1.5); this second index is important since an average Z score may mask significant areas of difficulties in a number of tasks (given a profile were some skills are good and others are impaired).

## 3. Data Analyses

We used two-tailed *t*-tests to compare the demographics and metabolic measures of the Italian and English AwPKU, as well as to compare the demographics and cognitive performance of each of these clinical groups with the corresponding control group. This allowed us to assess whether the Italian AwPKU demonstrated similar impairments to the English AwPKU [15]. Additionally, we computed Z scores for each PKU participant and each task, using the corresponding control group (subtracting each test value from the corresponding control mean and dividing by the control SD). Average Z scores for Italian and English AwPKU were analyzed using two-tailed *t*-tests to compare size of standardized effects. For this and all other analyses in the paper, we were more interested in comparing patterns across groups than in the significance of an individual measure. However, with the *t*-tests, we have indicated comparisons which remained significant after a Bonferroni correction.

To demonstrate that our battery is as sensitive to impairment in Italian as in English, we carried out Person r correlations between all our cognitive measures and measures of metabolic control, both current and historical. This resulted in a high overall number of correlations for each clinical group (*N* = 144). Individually, each correlation is not very reliable since correlations are notoriously unstable with a small N (19 participants). However, we were not interested in the significance of individual correlations, but, instead, in establishing the sensitive of our tasks to metabolic controls in Italian as in English, across measures. Considering patterns across a large number of cognitive tasks and metabolic measures will reduce error and boost power. We used a χ^2^ s to assess the significance of a positive/negative correlations ratio against a chance 50/50 ratio, which is expected if no true relation exists between cognitive performance and metabolic control. In addition, we used a one-sample *t*-test to assess whether the average correlation was significantly different from 0 (see Romani et al., [6] for a similar methodology). We carried out these analyses on both Italian and English PKU samples.

## 4. Results

### 4.1. Demographics

Demographic and metabolic information for English and Italian PKU groups are shown in Table 1. The Italian and English groups were matched for age, education, and gender. They did not differ significantly for of any of these variables or for IQ. However, the average Phe level was higher in the Italian group, with differences reaching significance for all age bands but adolescence. In addition, Phe levels were more variable in the Italian than in the English group in adulthood and when measured throughout the lifetime.

### 4.2. Cognitive Performance

#### 4.2.1. AwPKU vs. Controls

Table 2 shows the performance of the Italian and English PKU groups compared to matched samples of healthy controls. Results show that both Italian and English AwPKU were impaired in a similar range of tasks with good overlap with the English PKU group.

Both Italian and English AwPKU were impaired in IQ measures and showed a reduced speed in allocating visuo-spatial attention and good accuracy in visual search tasks. Both groups showed impairments in tasks tapping visuo-motor coordination (but for the Italian group differences reached significance only for the digit symbol test). The Italian group also showed impairments in the TMT b and a-b and in the fluency tasks and in the RVP task. Both groups showed no difference in the WCST. Finally, both groups performed well in verbal and visuo-spatial short-term memory tasks (digit span, Corsi Block tapping test) and in tasks tapping learning and memory with only a marginal impairment for the Italian AwPKU.

#### 4.2.2. Italian AwPKU vs. English AwPKU

Table 3 compares standardized performance of Italian and English PKU groups. Despite their worse metabolic control, the Italian group performed significantly worse than the English PKU group in only a few tasks. They were worse on the TMT B and B-A and on letter-fluency, marginally worse on list recall after interference (retention A6), but better on the Corsi Block span. Overall, their performance was numerically worse in terms of average Z score, but this was not statistically significant. When we co-varied concurrent Phe, only the differences in trail making test B-A and Corsi span remained significant (*p* = 0.009 and *p* < 0.001); differences in Trail making B and letter fluency were only marginally significant (*p* = 0.06 and *p* = 0.08).

A detailed comparison of results in visual search tasks highlights the similarity of patterns in the Italian and English PKU groups. Figure 1 shows RTs in the different conditions. Errors were too few for analysis. Across groups, RTs in feature search do not increase with the number of distractors. This is because in this condition the target item “pops out” and a parallel search suffices for a correct answer (generating a flat profile). Instead, across groups, there is a steep increase in RTs with the number of distractors in conjunction search. This is consistent with the need to serially explore the distinct locations of the display in this condition. Moreover, all groups show slower RTs with “No” rather than “Yes” trials, especially in the conjunction search. This is also expected. To decide that an item is not present you need to check all locations in the display; instead to find a target item, only half of the locations need to be checked on average. The PKU groups show the same patterns of the controls, but they are slower in all conditions.

Trials (F (1, 36) = 43.9; *p* < 0.001; η_p_^2^ = 0.550). There were also a number of significant interactions: task x distracters (F (2, 72) = 77.7; *p* < 0.001; η_p_^2^ = 0.683), because responses were slower with more distracters in the conjunction search, but not in the feature search; distracters x trial (F (2, 72) = 16.0; *p* < 0.001; η_p_^2^ = 0.307) because the effect of the number of distracters was more marked in the “no” than the “yes” trials; task x trial (F (1, 36) = 64.8; *p* < 0.001; η_p_^2^ = 0.643) because slower responses in the “no” rather than the “yes” trials occurred in the conjunction search, but not in the feature search. Finally, there was a three-way interaction: task x distracters x trial (F (2, 72) = 12.6; *p* < 0.001; η_p_^2^ = 0.259) because responses were slower with the number of distracters more in the “no” than the “yes” trials, but only in the conjunction search. Crucially, however, there were no significant interactions within the group.

When PKU participants are compared to controls, differences are constant in all conditions, except in the conjoined task where the English PKU group shows increasing differences from controls with number of distractors: a fanning-out pattern (distracters x group: F (2, 72) = 3.8; *p* = 0.03; η_p_^2^). Instead, differences from controls always remained stable in the Italian group (F (2, 72) = 0.25; *p*= 0.78). We will comment on this difference in the General Discussion.

Finally, we want to highlight that, as reported for the English sample (Palermo et al., 2017), the variability in cognitive performance in Italian PKU participants is striking. Five PKU participants (26% of the sample) had a normal cognitive profile when compared to the control group (average Z score: −0.2; −0.7−0.1; % z score =>1.5: 5.0; 0–8.3; expected 6.7%; Full IQ = 114; 104–124). These participants had fast speed of processing while maintaining a good accuracy.

### 4.3. Cognitive Outcomes in Relation to Metabolic Control

We ran Bivariate Pearson r correlations between metabolic measures taken at different times during the life span and our adult cognitive measures. To reduce the number of variables per task, we only run correlations for search tasks, with RT measures; for TMT, with the B–A condition; for the WCST, with the total errors; and for the Rey, with total errors over 1–5 trials and with delay recall. Table 4 shows results for the Italian sample. Results for the larger sample of English AwPKU are reported in Romani et al. [5]. We do not report correlations with the English matched sub-sample because correlations with a small sample are notoriously unstable. Thus, results with a smaller sub-sample may differ from the results obtained with a larger sample in potentially misleading ways. However, to compare results for the two languages with samples of an equal size, we also report, with both PKU samples, the % of positive correlations and the average correlation.

The Italian sample had widespread significant correlations with only a few tasks failing to show any significant correlation with any metabolic measure (pegboard task, the semantic fluency task, and the Rey test). Almost all correlation (91%) were positive (132/144; χ^2^ = 60.5; *p* <.001). The average correlation across tasks was 0.29 (SD = 0.22), which was significantly different from 0 (one sample *t*-test (142) = 17.0; *p* < 0.001). In the English matched sample (*N* = 19), 69% of correlations were positive (99/144 = 69%; χ^2^ = 10.5; *p* < 0.001) and the average correlation was 0.12 (SD = 0.25; one sample *t*-test (142) = 5.7; *p* < 0.001). Taken together, these results indicate that our tests are sensitive to the level of metabolic control in both languages.

Considering the qualitative pattern of correlations in the Italian sample, a number of significant features can be noted which replicate findings from the larger English sample [5]: 1. Adult cognitive performance correlates with current Phe levels, but also with historical Phe records with significant correlations across the life-span; 2. Cognitive performance correlates with metabolic measures in terms of both average Phe levels and Phe fluctuations (average SD per year), this confirms the importance of maintaining not only low average Phe levels, but also of being constant and of avoid Phe peaks; 3. There are interactions between type of function an age when metabolic control is measured: some functions are more effected by childhood Phe measures than by current Phe levels (see visuo-attentional speed). Other functions, instead, show as much of an association with current levels as with historical measures (IQ, visuo-motor control, sustained attention, memory, and learning.

## 5. Discussion

There is a lack of studies which have assessed the behavioural effects of metabolic control in individuals with PKU across countries with the same testing materials, comparing sensitivity. This, instead, is important for establishing common PKU testing batteries and to accrue results across centers for a rare disease. In spite of tremendous advances in our knowledge of PKU, we lack a complete and reliable picture of cognitive outcomes in relation to metabolic controls at different ages, which is crucial to establish the efficacy of new therapeutic interventions and to track developmental trends which may demonstrate either cognitive improvements (by bridging developmental gaps) or cognitive deterioration with age (due to abandoning the diet and/or accelerated aging). Our study tested cognitive outcomes in two samples of Italian and English AwPKU closely matched for gender, age, and education, using the same or closely matched materials.

### 5.1. Metabolic Control

Our results showed better metabolic control in the English than the Italian PKU sample. This could be due to differences in clinical practice (the English group is one of the better controlled groups described in the literature so far, which is likely due to the strong clinical advice received) or to cultural differences in approaches to food across countries. While in Italy, the diet may be less protein-based than in England, the stronger centrality of food in Italian society may make following a separate diet more difficult. Our matched samples are small, and we do not have corroborative information indicating whether following a PKU diet may be easier in England than in Italy. Nonetheless, our results point to variability in metabolic control across groups, suggesting the need to explore possible, modulating socio-cultural factors which could affect clinical management.

### 5.2. Cognitive Impairments

Our results showed similar patterns of cognitive outcomes and associations with metabolic measures across the Italian and English groups. The Italian PKU participants performed worse than the matched English PKU participants in a few tasks, consistent with their worse metabolic control, but the overall level of impairment and the pattern of spared and impaired functions were similar across groups. The results with an Italian sample confirm patterns of impaired and spared functions previously reported with English participants [15]. They confirm substantial impairments in the following:
-Speed of processing, at least, with tasks tapping visual attention (no other RT tasks were used), confirming previous results [11,27], but good performance in the same tasks in terms of accuracy, confirming that AwPKU are slow, but accurate.-Executive functions in terms of flexibility and planning (TMT B and B–A and verbal fluency), but no impairment in the WCST (see also Moyle et al. [49] for impairments in fluency, but no impairments in the TMT B; see also Brumm et al. [25]; Smith et al. [50], for impairments in the WCST; see Ris et al. [51], and Channon et al. [52], for impairments in a test similar to the WCST).-Sustained attention (Rapid Visual Information Processing task; see also Channon et al. [52]; Bik-Multanowski et al., [53]; Jahja et al. [27]; Schmidt at al. [47]; Weglage et al. [9]).-Visuo-motor coordination (marginal impairment in the peg-board task; see also Griffiths et al. [54]; Pietz et al. [55], but also Brumm et al. [25], for no impairment).


The Italian sample showed no impairments in verbal and visual short-term memory. Performance was in fact better than the controls in the Corsi task, confirming the lack of impairment found in Palermo et al., [15]. We have previously found a marginal impairment in digit span. The present results are also consistent with Moyle et al., [49], who found no impairment in the working memory index of the WAIS, and with Brumm et al. [25], who found no impairment in the forward digit span, but an impairment with the backwards span (see also Jahja et al. [27] for deficits with increasing working memory load).

Finally, our results confirm no or marginal impairments in verbal and visual learning (for no impairments in adults see also Channon et al. [52]; Moyle et al. [49]; for a review in children showing mixed results see Janzen and Nguyen, [14]). These results support the idea that AwPKU perform better when they can rely on learning and stored knowledge (see good performance in word spelling; better word than nonword reading speed; and better digit span than nonword repetition; [15,16]).

When we have analysed patterns in visual search tasks, the Italian and English PKU samples also showed similar results. Both groups were slower in conjunction than in feature search tasks and showed increased delays with a number of distractors, especially in conjoined search and target absent conditions. Compared to matched control, both Italian and English PKU samples showed delays across tasks and conditions. However, in conjunction search, the English PKU group showed increased delays with a number of distractors (a fanning out pattern), while the Italian PKU group showed stable differences. In Romani et al. [16], we argued that two different types of difficulties may contribute to a speed reduction in AwPKU: 1. Some specific processing difficulties such as a difficulty in allocating visual attention; this will result in increasing differences from the control with number of distractors; and 2. A general tendency to be more cautious/tentative in returning an answer which will result in a fixed delay across conditions (a flat profile across conditions; a pattern that we have seen with language tasks, see Romani et al., [16]). The Italian PKU sample is relatively small, and it is possible that noise may have obliterated a fanning-out pattern in conjunction search. With this caveat, a stable difference from controls, which is not affected by task difficulty, is more consistent with a delay in making decisions than with processing difficulties.

Taken together, our results highlight how PKU does not impact cognition homogeneously. Some functions are spared while others—like visuo-attentional speed—are severely impaired, showing average Z scores between 1.2 and 1.5 from the control mean (and Z score SDs between 1.2 and 2.3), which indicate that some people with PKU are at the extreme margin of the normal distribution. To consider this variability is important for an accurate evaluation of outcomes in relation to current and future treatment options. Our results also confirm the extreme variability in performance across individual participants. In the Italian sample, 5 participants (26% of sample) showed a completely normal cognitive performance.

Concerning the relationship between cognitive and metabolic measures, our Italian results confirm extensive correlations between current and historical Phe measures and performance in cognitive tasks. Previous studies have also found correlations between current and life-time Phe levels and cognitive functions (see Brumm et al. [25] for correlations with backwards span, fluency, WCST; Smith et al. [50] for a correlation with WCST; Moyle et al. [49] for correlations with TMT and visual memory; Jahja et al. [27] for correlation with visual search tasks with a memory load; see Romani et al. [5], and Hofman et al. [56], for a more detailed review of the literature). Other studies have found differences in cognitive abilities between AwPKU with better versus worse metabolic control. For example, Nardecchia et al. [26] found that patients with a worse metabolic control had a lower IQ and worse performance at the WCST and at the Elithorn’s Perceptual Maze Test; Bik-Multanowski et al. [53] found that patients with a worse metabolic control performed worse in RVP, Spatial Span (SSP), Spatial Working Memory (SWM) and Stop Signal Task (SST; see also Brumm et al. [25] and Palermo et al. [15] for results on more comprehensive sets of cognitive tasks).

Our results also confirm that both measures in terms of average Phe and Phe fluctuations correlate with performance [57] and that there are interactions between type of function and age of metabolic control. Some functions are more affected by historical Phe and others by current Phe. Speed in visual attentional tasks is associated mainly with childhood Phe (both average and SD), while tasks tapping visuo-motor coordination (digit-symbol, TMT), sustained attention and memory and learning are also affected by current Phe [5,11].

Our results highlight the importance of using the right cognitive and metabolic measures to assess outcomes. For example, a recent study by Bartus et al. [58] found no correlations between Phe levels (current and life-time) and performance on three tasks of the computerized Cambridge Cognition Test (CANTAB)—Motor screening test, Spatial Working Memory test, and The Stoking of Cambridge—in 47 AwPKU, nor any difference between groups with high vs. low Phe levels. However, we also found no correlations when we used similar tasks with our Italian/English samples (motor speed tapped by simple RT, visual WM tapped by Corsi span) or with larger English sample (the Tower of Hanoi was not tested here; see Romani et al., [5]). This example shows the importance of using both a comprehensive and an ad hoc set of tasks when assessing outcomes and their relationship with metabolic variables. Failing to do so runs the risk of reaching the wrong conclusions regarding the effects of relaxing metabolic control.

## 6. Conclusions

It is important to track the cognitive performance of people with PKU across the lifespan. This is the first generation of early-treated AwPKU to move towards middle-age and people with PKU show a tendency to progressively relax the diet with age [55,59,60,61]. However, we do not know what effects prolonged high-levels of Phe may have on aging brains. Our results suggest that the effect of Phe on the brain is different in childhood to early adulthood [6], but further interactions may be seen later in life. Knowing which functions and relative tasks are most affected and most sensitive to Phe in young adults with PKU provides an important base-line to compare outcomes across the life-span and evaluate the effectiveness of treatment. Our study has contributed to an identification of sensitive tasks by showing consistency across Italian and English samples in the patterns of impaired and spared functions and similar patterns of correlations with metabolic measures and by replicating previous findings. This provides preliminary evidence that common PKU batteries can be used across countries to detect impairments with similar sensitivity. The similarity of the results across the Italian and English PKU samples justified combining results in a single database in a follow-up study, giving us more power to assess the interactions between types of metabolic variable (Phe average vs. Phe fluctuation), age of metabolic control (childhood, adulthood, current) and type of cognitive functions, and more power to assess the relationships between cognitive scores and adherence to metabolic guidelines [6].

### Limitations

The main limitation of our study lies in the small sample of AwPKU tested in Italian. Although this small sample is still sufficient to establish that our tasks are sensitive to metabolic control, larger samples are necessary to compare individual correlations and replicate our preliminary findings that qualitive patterns are the same across languages and nationalities. There was variability in the correlation patterns shown by the Italian and English PKU samples. This is not surprising. The two PKU samples were not matched for metabolic controls. More importantly, correlations are notoriously unstable and have small samples, and this is a major stumbling block for research trying to establish the relationship between metabolic controls and cognitive outcomes in people with PKU. However, it is precisely this limitation that makes it important to establish that tasks are equally sensitive across different nationalities so that results can be accrued. We hope that that our study will be followed by further research to assess the sensitivity of the same tasks across languages in people with PKU.

## Figures and Tables

**Figure 1 nutrients-12-03033-f001:**
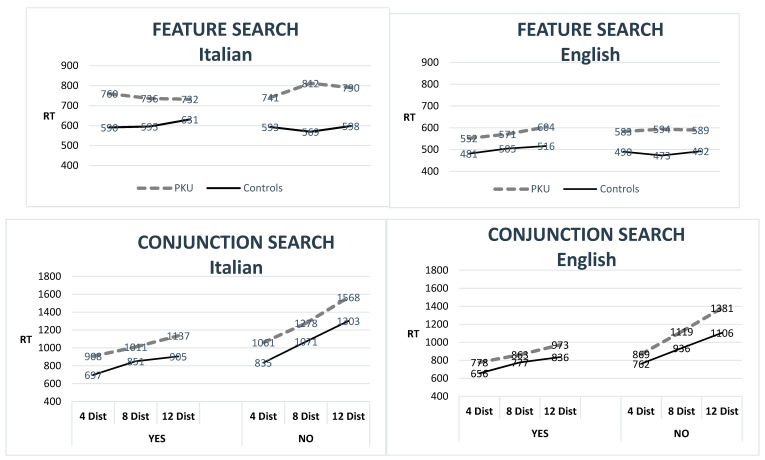
Performance of the UK and Italian PKU groups and relative control groups on the visual search tasks. Dist: distractors, PKU: Phenylketonuria, RT: reaction Times.

**Table 1 nutrients-12-03033-t001:** Demographic Variables in Terms of Age, Gender, Years of Education and Metabolic Control across Ages for the Two Matched Groups of Italian and English AwPKU. Blood Phe measured in μmol/L.

	Italian AwPKU		English AwPKU		English vs. Italian
	*N* = 19		*N* = 19		*p* Value
	Mean	SD	Mean	SD	
**Age**	25.4 (range: 19–33)	*4.1*	25.3 (range: 18–36)	*6.1*	n.s.
**Education (in years)**	14	*1.8*	14.6	*1.9*	n.s.
**Gender (M/F)**	8//11		8//11		
**Verbal IQ**	98.8	*12.9*	100.4	*8.9*	n.s.
**Performance IQ**	99.3	*15*	103.3	*12.9*	n.s.
**Full IQ**	98.9	*14.5*	102.1	*10.4*	n.s.
**Childhood (0–10 years)**					
Phe Average Median	499	*149*	386	*168*	*p* = 0.04
Phe Fluctuation	227	*63*	198	*50*	n.s.
Mean N observationsper participant	208	*79*	259	*156*	n.s.
**Adolescence (11–16 years)**					
Phe Average Median	702	*194*	612	*293*	n.s.
Phe Fluctuation	170	*51*	165	*34*	n.s.
Mean N obs. per participant	77	*54*	98	*74*	n.s.
**Adulthood (17 years +)**					
Phe Average Median	970	*239*	733	*344*	*p* = 0.02
Phe Variation	217	*65*	122	*41*	*p* < 0.001
Mean N obs. per participant	58	*49*	62	*58*	n.s.
**Lifetime**					
Phe Average Median	695	*198*	516	*233*	*p* = 0.02
Phe Fluctuation (SD)	208	*46*	171	*33*	*p* < 0.01
Mean N observations. per participant	344	*149*	419	*232*	n.s.
**Current Phe**	1042	*428*	677	*382*	*p* = 0.01
**Range**	454–2081		65–1465		

AwPKU: adults with PKU, IQ: intelligence quotient, n.s.: not significant, Phe: Phenylalanine.

**Table 2 nutrients-12-03033-t002:** Cognitive Performance of Italian and English PKU Participants and Healthy Controls Matched for Age and Educations to the Respective Clinical Groups.

	ITALIAN PARTICIPANTS (*N* = 19 In Each Group)					ENGLISH PARTICIPANTS (*N* = 19 In Each Group)				
	AwPKU		Controls		Diff.	AwPKU		Controls		Diff.
	Mean	SD	Mean	SD	*p* Value	Mean	SD	Mean	SD	*p* Value
**Age**	25.4	*4.1*	24.7	*3.4*	n.s.	25.3	*6.1*	24.4	*5.35*	n.s.
**Education (in years)**	14.0	*1.8*	13.9	*1.7*	n.s.	14.6	*1.9*	14.8	*1.7*	n.s.
**Gender (M/F)**	8//11		8//11			7//12		6//13		
**VIQ**	98.8	*13*	108.5	*13.1*	0.03	100.4	*8.9*	109.6	*9.6*	<0.001 #
**PIQ**	99.3	*15*	107.1	*9.6*	0.07	103.3	*12.9*	110.2	*12.6*	n.s.
**FIQ**	98.9	*15*	110.1	*12.0*	0.01	102.1	*10.4*	111.1	*10.3*	0.01
**VISUAL ATTENTION**										
**Simple Detection** (RT—ms)	325	*46*	315	*44.6*	n.s.	331	*50.7*	304.9	*51.2*	0.06
**Detention with Distractors**										
RT—ms	483	*104*	430	*58*	0.02	433	*56.4*	392	*54.2*	0.03
% errors	0.8	*0.6*	0.5	*1.1*	n.s.	0.8	*0.8*	0.5	*0.6*	n.s.
**Feature Search**										
RT—ms	752	*262*	586	*113*	0.02	581	*111*	492.3	*70.5*	0.01
% errors	1.6	*4.7*	0.6	*1.2*	n.s.	1.9	*2.6*	2.3	*2.3*	n.s.
**Conjunction Search**										
RT—ms	1152	*237*	938	*203*	<0.001 #	998	*162*	846	*129.0*	<0.001 #
% errors	3.2	*4.7*	2.8	*4.7*	n.s.	2	*2.2*	3.5	*5.4*	n.s.
**VISUO-MOTOR COORDINATION**										
**Pegboard** (Time–s)	76.7	*11*	70.6	*9.0*	0.06	71.9	*9.9*	66.0	*6.1*	0.03
**Digit Symbol** (%errors in 90 s)	41.6	*12*	39.4	*9.1*	<0.001 #	37.1	*11.6*	27.9	*10.3*	0.01
**Trail-Making Test** A (Time–s)	32.4	*15*	29.9	*12.1*	n.s.	24.0	*7.0*	20.2	*3.7*	0.04
**EXECUTIVE FUNCTIONS**										
**WCST**										
Total errors	13.6	*6.9*	13.6	*5.3*	n.s.	13.8	*8.6*	11.9	*5.5*	n.s.
Perseverative responses	7.0	*3.1*	7.8	*3.3*	n.s.	7.9	*5.1*	7.5	*5.3*	n.s.
N of Completed Categories	4.0	*1.1*	4.1	*1.0*	n.s.	3.9	*1.2*	4.3	*0.9*	n.s.
**Trail-Making Test**										
B (sec)	79.8	*34*	54.4	*19.7*	0.01	42	*10.8*	43.2	*16.0*	n.s.
B–A (sec)	47.5	*29*	24.4	*13.1*	<0.001 #	18	*8.0*	22.9	*14.3*	n.s.
**Verbal Fluency**										
Letter (correct answers)	39.0	*11*	47.5	*9.3*	0.01	36.3	*11.6*	40.5	*15.0*	n.s.
Semantic (correct answers)	20.1	*4.8*	23.5	*3.6*	0.02	21.6	*5.8*	25.1	*5.0*	0.06
**Rey Auditory Verbal Learning**										
Retention after interference (% errors A6)	17.2	*13*	8.4	*7.6*	0.02	19.6	*18.3*	15.1	*16.9*	n.s.
**SUSTAINED ATTENTION**										
**RVP** (% of errors)	24.2	*12*	15.9	*9.6*	0.03	16.8	*9.5*	13.2	*9.4*	n.s.
**SHORT-TERM MEMORY**										
**Digit span**	5.9	*0.9*	5.9	*1.1*	n.s.	6.2	*0.8*	6.5	*0.9*	n.s.
**Corsi Block span**	5.9	*1.0*	5.0	*1.1*	0.01	5.3	*0.8*	5.6	*0.9*	n.s.
					(opposite to expected)					
**MEMORY and LEARNING**										
**Rey Auditory Verbal Learning**										
Trial A1–A5 (% errors)	20.8	*8.8*	19.7	*6.1*	n.s.	24.5	*11.4*	21.7	*9.3*	n.s.
Delayed Recall (% errors)	10.2	*11*	4.9	*7.0*	0.08	18.6	*18*	14.7	*14.8*	n.s.
**Paired Associate Visual learning** (% errors)	2.3	*2.2*	1.2	*1.3*	0.07	2.5	*3*	1.8	*1.8*	n.s.

Italian and English PKU groups are also matched a-priori for age and education. Diff. = Difference. # = comparison which remains significant after Bonferroni correction. WCST: Wisconsin Card Sorting Test, RT: reaction Times, RVP: Rapid Visual Information Processing task, VIQ: verbal IQ, PIQ: performance IQ, FIQ: full IQ.

**Table 3 nutrients-12-03033-t003:** Cognitive Performance of English and Italian PKU Groups in Term of Z Score Computed from Relative Control Groups Matched for Age, Gender, and Education.

	Italian AwPK Z Score		English AwPKU Z Score		Diff.
DOMAIN/TASK	Mean	SD	Mean	SD	*p*
**IQ**	1.0	*1.0*	0.9	*1.0*	n.s.
VISUO-SPATIAL ATTENTION RTs					
**Simple Detection**—ms	0.2	*1.0*	0.5	*1.0*	n.s.
**Detention with Distractors**—ms	1.3	*2.1*	0.8	*1.0*	n.s.
**Feature Search**—ms	1.5	*2.3*	1.3	*1.6*	n.s.
**Conjunction Search** ms	1.1	*1.2*	1.2	*1.2*	n.s.
VISUAL ATTENTION accuracy					
**Detention with Distractors**—% errors	0.2	*0.6*	0.5	*1.3*	n.s.
**Feature Search**—% errors	0.9	*4.0*	−0.2	*1.1*	n.s.
**Conjunction Search**—% errors	0.1	*1.0*	−0.3	*0.4*	n.s.
VISUO-MOTOR COORDINATION					
**Pegboard**—s	0.7	*1.2*	1.0	*1.6*	n.s.
**Digit Symbol**—% errors in 90 s	0.2	*1.3*	0.9	*1.1*	n.s.
**Trail-Making Test A**—s	0.2	*1.2*	1.0	*1.9*	n.s.
**EXECUTIVE FUNCTIONS**					
**WCST**					
Total errors	0.0	*1.3*	0.3	*1.5*	n.s.
Perseverative responses	−0.3	*0.8*	0.1	*1.0*	n.s.
N of Completed Categories	0.1	*1.0*	0.3	*1.3*	n.s.
**Trail-Making Tests**					
B (s)	1.3	*1.7*	−0.1	*0.7*	<0.001#
B–A (s)	1.8	*2.2*	−0.3	*0.6*	<0.001#
**Verbal Fluency**					
Letter (correct answers)	0.9	*1.1*	0.3	*0.8*	0.05
Semantic (correct answers)	1.0	*1.3*	0.7	*1.2*	n.s.
**Rey Auditory Verbal Learning**					
Retention after interference (% errors A6)	1.1	*1.7*	0.3	*1.0*	0.07
**SUSTAINED ATTENTION**					
**RVP** (% of errors)	0.9	*1.3*	0.4	*1.0*	n.s.
**SHORT-TERM MEMORY**					
**Digit span**	0.0	*0.8*	0.3	*0.9*	n.s.
**Corsi Block span**	−0.8	*0.9*	0.3	*0.9*	<0.001#
**LEARNING**					
**Rey Auditory Verbal Learning Test**					
Trial A1–A5—% errors	0.2	*1.4*	0.3	*1.2*	n.s.
Delayed Recall—% errors	0.8	*1.5*	0.3	*1.2*	n.s.
**Paired Associate Visual learning**—% err	0.9	*1.7*	0.4	*1.7*	n.s.
**OVERALL Z SCORE** excluding IQ					
mean	0.59		0.42		n.s.
SD	0.76		0.58		

Diff. = Difference. # = comparison which remains significant after Bonferroni correction.

**Table 4 nutrients-12-03033-t004:** Correlations Between Phe Measures Across the Life-Span and Cognitive Performance in Different Domains for the Italian PKU Sample (*N* = 19).

		VISUAL ATTENTION SPEED				VISUO-MOTOR		EF-MONITORING					SUST.	LEARNING AND MEMORY		
ITALIAN PKU	FSIQ	Simple	Detection	Feature	Conj.	Peg-Board	Digit	WCST	TMT	Digit	Corsi	Sem	ATTENT	Rey	Rey	Paired Ass.
PHE		RT	With distractors RT	Search RT	Search RT	Sec.	Symbol	Total err	b–a	Span	Span	Fluency	RVP %	Words a1–a5	Words Delayed	Visual Learning
**0–10 yrs**																
Average	0.38	0.43	0.42	0.15	0.30	0.05	0.28	0.19	**0.55** *	**0.57** *	0.37	0.03	0.43	0.17	0.08	0.30
SD	0.31	**0.73** **	**0.73** **	0.34	**0.49** *	0.02	0.13	−0.10	**0.55** *	**0.66** **	0.42	0.08	**0.57** *	0.34	0.25	0.36
**11–16 yrs**																
Average	**0.53** *	0.08	0.30	0.41	0.35	0.18	0.38	0.28	0.44	0.32	0.29	0.41	0.63 **	0.14	0.08	**0.49** *
SD	0.02	0.29	0.32	0.37	0.20	−0.25	−0.10	−0.08	0.20	0.16	−0.04	0.38	0.37	−0.18	−0.29	0.30
**17 yr to now**																
Average	**0.58** **	0.03	0.37	0.38	0.41	0.32	**0.60** **	**0.59** **	**0.56** *	**0.60** **	**0.60** **	0.24	**0.48** *	0.11	0.09	0.30
SD	**0.52** *	−0.06	0.06	0.16	0.22	0.29	0.43	**0.46** *	**0.52** *	0.38	0.28	0.21	0.33	0.05	−0.11	−0.20
**Current**	0.41	−0.10	0.05	0.03	0.01	0.12	**0.61** **	**0.58** **	**0.54** *	**0.60** **	0.37	0.00	0.30	0.21	0.16	0.11
**Lifetime**																
Average	**0.55** *	0.23	0.42	0.29	0.39	0.03	0.37	0.31	0.55 *	**0.58** **	0.33	0.23	**0.58** *	0.12	0.12	0.38
SD	**0.47** *	0.45	**0.54** *	0.40	0.45	0.05	0.24	0.16	**0.60** *	**0.55** *	0.35	0.28	**0.58** *	0.13	0.00	0.26

Highlighted rows compare childhood Phe measures with current Phe. To facilitate interpretation, positive correlations always indicate that high Phe was associated with a worse performance. Thus, for IQ, digit span, Corsi span, and semantic fluency correlations were reversed. Significant measures are in bold. * = significant < 0.05; ** significant < 0.01. Phe SD: Phe fluctuations, FSIQ: full scale IQ, TMT: Trail Making Test, ATTENT: attention, PHE: Phenylalanine.

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
