# Peer review of "Cognitive Outcomes and Relationships with Phenylalanine in Phenylketonuria: A Comparison between Italian and English Adult Samples"

_nutrients, 2020, doi:10.3390/nu12103033_

Round 1

Reviewer 1 Report

The paper by Romani et al is a resubmission of an important area of study in the context of PKU and cognitive outcomes. Cognitive outcomes in relation to metabolic control remains to be fully understood, especially from a multi-country perspective. Comparing two nations, the present manuscript tested cognitive outcomes in Italian and English adults with PKU. While it is difficult to standardise research methods in multi-country trials, I would like to reiterate that the author’s should be commended for closely matching cohorts according to gender, age and education. The manuscript is much improved. I would like to thank the author’s for adhering to the MDPI formatting guidelines and correcting the issues with line numbers as it made the resubmission easier to review and provide. The referencing style, however, is intermittent with Arabic numbers in places and author name/year in others. Be consistent throughout. I would focus on the tense used throughout the manuscript. Consider when present and past tense should be use.

Abstract

Objective is clear.

Line 31: Write phenylalanine in full prior to abbreviating.

Introduction

Overall, the introduction is remains long and difficult to follow in places. I would encourage condensing if possible and focusing only what is relevant to the need for the submitted study

Line 115-127: Make this paragraph past tense as the trial has been completed… ‘we conducted’ and opposed to ‘we will’.

Methods

2.1 Participants

Thank you for providing detail about how the cohorts were selected and matched and adding detail about the sample size estimate.

2.2. Metabolic measures

What is the intraclass correlations on the labs that quantified sample? Do you have this information available?

2.3. Cognitive assessment

There remains a lot of information in this section and while this is important for reader repeatability, could the authors consider condensing this section so it is easier to follow. The authors could consider supplementary material.

Thanks for adding a statistical analysis section.

Remove the present tense in sentences justifying the use of particular analysis techniques. Instead consider expanding sentences such as below.

From ‘Average Z-scores for Italian and English AwPKU were then compared using two-tailed t-tests. This allowed us to compare size of standardized effects.’ this to this ‘Average Z-scores for Italian and English AwPKU were compared using two-tailed t-tests to compare size of standardized effects.’

Results

4.1 Demographics

Tense in the second sentence is in incorrect. Use did instead of do.

Figure 1: What does RT denote? Please make this clear.

General

The referencing style does not reflect that permitted by Nutrients. References must be numbered in order of appearance in the text

Author Response

Dear reviewer, thank you very much for your very helpful comments.  The paper has improved thanks to them.  I hope you find our further minor revisions satisfactory.  Best regards,

Cristina Romani

(for all the others)

The paper by Romani et al is a resubmission of an important area of study in the context of PKU and cognitive outcomes. Cognitive outcomes in relation to metabolic control remains to be fully understood, especially from a multi-country perspective. Comparing two nations, the present manuscript tested cognitive outcomes in Italian and English adults with PKU. While it is difficult to standardise research methods in multi-country trials, I would like to reiterate that the author’s should be commended for closely matching cohorts according to gender, age and education. The manuscript is much improved. I would like to thank the author’s for adhering to the MDPI formatting guidelines and correcting the issues with line numbers as it made the resubmission easier to review and provide. The referencing style, however, is intermittent with Arabic numbers in places and author name/year in others. Be consistent throughout. I would focus on the tense used throughout the manuscript. Consider when present and past tense should be use.

Abstract

Objective is clear. Thank you

Line 31: Write phenylalanine in full prior to abbreviating.  Yes done.

Introduction

Overall, the introduction is remains long and difficult to follow in places. I would encourage condensing if possible and focusing only what is relevant to the need for the submitted study.  We have worked on the Introduction.  We have streamlined it and made it more coherent.

Line 115-127: Make this paragraph past tense as the trial has been completed… ‘we conducted’ and opposed to ‘we will’.  Yes done.

Methods

2.1 Participants

Thank you for providing detail about how the cohorts were selected and matched and adding detail about the sample size estimate. Thank you

2.2. Metabolic measures

What is the intraclass correlations on the labs that quantified sample? Do you have this information available?  Unfortunately, this information is not available

2.3. Cognitive assessment

There remains a lot of information in this section and while this is important for reader repeatability, could the authors consider condensing this section so it is easier to follow. The authors could consider supplementary material.  Since this study is about testing materials we feel that some information about the tests should be in the main text of the paper.  We have already condensed test descriptions.

Thanks for adding a statistical analysis section. Thank you

Remove the present tense in sentences justifying the use of particular analysis techniques. Instead consider expanding sentences such as below.  Present and future tenses were changed with past tenses.

From ‘Average Z-scores for Italian and English AwPKU were then compared using two-tailed t-tests. This allowed us to compare size of standardized effects.’ this to this ‘Average Z-scores for Italian and English AwPKU were compared using two-tailed t-tests to compare size of standardized effects.’  Thank you this has been changed as suggested.

Results

4.1 Demographics

Tense in the second sentence is in incorrect. Use did instead of do.  We changed the tense in the whole paragraph.

Figure 1: What does RT denote? Please make this clear.  We have specified that RTs refer to ‘reaction times’ the first time we used ‘RTs’.

General

The referencing style does not reflect that permitted by Nutrients. References must be numbered in order of appearance in the text. Pease note that some references have been changed into number by the journal.  Hopefully, references will be made uniform in the proofs.  

Reviewer 2 Report

Comments and suggestion for the authors

I think this is a very interesting and difficult study, because there are many variables to consider.

General

The objective of the study to assessed if the same cognitive batteries can be used cross-nationally in Pku patients is very interesting.

I think it is necessary to consider in this work that in the Italian group 4 patients had a diagnostic blood phe levels between 600 and 1200mcmol/ml unlike the English group where every patients had  phe>1200mcmol/ml, also in the Italian group there were 3 patients with sapropterine treatment. For these circumstances the Phe average median in the Italian group could be even higher and this is important to take it into account

I think it could be also interesting to consider the tyrosine levels if you have them.

Introduction: I think it is too long, and when studies are cited, I think that putting one or two authors is enough, the rest can be added in the bibliography

Line 43: Asbjorn Folling was Norwegian.

line 46:  countries instead of counties

line 182 explain the acronym FSIQ

Method:

Participants

Very important the characteristic of the patients

Metabolic measures:

Thank you for providing details about phe measurements conditions and the phe fluctuations calculation.

Cognitive assessment

I think is too long. Maybe the authors can consider not to explain in detail the test

Results

Demographics

 Table 1 explain the acronym VIQ PIQ, FIQ under the table

Cognitive performance

It is difficult to find the table 2 epigraph, maybe the author could consider put it independently

Author Response

Dear reviewer, thank you very much for your very helpful comments.  The paper has improved thanks to them.  I hope you find our further minor revisions satisfactory.  Best regards,

Cristina Romani

(for all the others)

I think this is a very interesting and difficult study, because there are many variables to consider.

General

The objective of the study to assessed if the same cognitive batteries can be used cross-nationally in Pku patients is very interesting.

I think it is necessary to consider in this work that in the Italian group 4 patients had a diagnostic blood phe levels between 600 and 1200mcmol/ml unlike the English group where every patients had  phe>1200mcmol/ml, also in the Italian group there were 3 patients with sapropterine treatment. For these circumstances the Phe average median in the Italian group could be even higher and this is important to take it into account. We agree with this observation. 

I think it could be also interesting to consider the tyrosine levels if you have them.  We had tyrosine levels but since some participants took supplements and others did not, they were too difficult to compare.

Introduction: I think it is too long, and when studies are cited, I think that putting one or two authors is enough, the rest can be added in the bibliography.  We have further reduced and streamlined the introduction and amended references only citing first authors when studies have more than two authors.

Line 43: Asbjorn Folling was Norwegian.  Sorry!! We corrected it.

line 46:  countries instead of counties - Corrected

line 182 explain the acronym FSIQ – We explained it

Method:

Participants

Very important the characteristic of the patients

Metabolic measures:

Thank you for providing details about phe measurements conditions and the phe fluctuations calculation.  Thank you!

Cognitive assessment

I think is too long. Maybe the authors can consider not to explain in detail the test.  We understand this comment.  However, we are reluctant to reduce our test descriptions further since the nature of the tests is central to the study.  Many PKU professions may not be familiar with the tasks we used.

Results

Demographics

 Table 1 explain the acronym VIQ PIQ, FIQ under the table.  We have now written in full: Verbal IQ, Performance IQ and Full IQ.

Cognitive performance

It is difficult to find the table 2 epigraph, maybe the author could consider put it independently.  Something was previously wrong with the title of this table; it has been corrected.

This manuscript is a resubmission of an earlier submission. The following is a list of the peer review reports and author responses from that submission.

Round 1

Reviewer 1 Report

Comparing two nations, the present manuscript tested cognitive outcomes in Italian and English adults with PKU. This is an important area of study in the context of PKU as cognitive outcomes in relation to metabolic control remains to be fully understood, especially from a multi-country perspective. While it is difficult to standardise research methods in multi-country trials, the author’s should be commended for closely matching cohorts according to gender, age and education. It was however very difficult to provide peer review as the submitted manuscript does not follow MDPI formatting guidelines. Line numbers were inconsistent; only starting at page 8 (section 3.2), citations were incorrect and revision of language is encouraged.

While there is merit in the submission I feel it will be hard for readers to follow in its current state. By using the statement ‘recruitment stopped when enough participants were tested’ it seems recruitment was stopped, and findings submitted when it suited the authors intention. There is no mention of sample size calculations to facilitate whether the study was sufficiently powered. The manuscript could be condensed in placed and some key information is missing.

Title

While the use of well-known abbreviations is permitted in scientific write up I would argue that the Phe is less known compared to PKU. I would therefore encourage writing phe in full and abbreviating phenylketonuria.

Telephone for corresponding should start +44 (0) 121…

Abstract

Objective

Objective is clear.

Methods

‘From our original battery, we selected a smaller set of tasks that was still large and comprehensive enough to cover’. Please reword the first part of the sentence that introduces your methods. At present the text is rather informal.

‘Tasks were closely matched in English and Italian’. Do the authors mean language? If so be clear.

Results

‘The Italian group had slightly worse metabolic control but showed levels of performance and patterns of impairment similar to the English group.’ I imagine the authors are referring to AwPKU here? Please ensure this is clear as it is difficult to know whether you mean the Italian and English group combined, AwPKU only or controls only.

As this is likely cited in the text the reference is not necessary remove.

General comments on abstract

As per the instructions for authors I would encourage the headings are removed from the abstract.

The terms English, Italian, matched and same are in italics in places but this is not consistent throughout the manuscript. Please ensure consistency throughout.

Write phenylalanine in full prior to abbreviating.

Introduction

Paragraph 1.

Although the authors direct the readers to reviews regarding IQ and cognitive outcomes it would be good to offer a brief overview to provide context to readers of these findings.

While the authors provide details concerning average Phe levels in AwPKU most of the literature cited is 20+ years old. I would encourage citing more contemporary data that reflects the countries being studied in the manuscript. For example, Green at al recently showed average phe levels for UK patients which differed according to dietary adherence.

Points 1-3 need to be re-formatted.

Paragraph 3

A space needs to be added between Phelevels.

Paragraph 4

When talking about negative effects of PKU on cognition why are some words underlined? This is not needed.

Overall, the introduction is long and difficult to follow. I would encourage condensing and focusing only what is relevant to the need for the submitted study.

Methods

2.1 Participants

Present the time for diagnosis for both countries in days or hours for constancy.

Thank you for providing detail about how the cohorts were selected and matched.

Was this research facilitated by a power calculation to identify necessary cohort numbers? The statement. From the statement ‘Recruitment stopped when enough participants were tested’ it seems recruitment was stopped, and findings submitted when it suited the authors intention.

2.2. Metabolic measures

Although details are provided about metabolic measures, there is no mention of how blood samples were taken. Was this the same method across both cohorts? What is the intraclass correlations on the labs that quantified sample? Typically, how many samples were averaged across the three age bands? If numbers are hugely different this may impact findings.

2.3. Cognitive assessment

As IQ quantification differed across cohorts, have both IQ methods been validated and are the tools comparable? If so mention this and if not add this to limitations.

There is a lot of information in this section and while this is important for reader repeatability, could the authors consider condensing this section so it is easier to follow. The authors could consider supplementary material.

There is no statistical analysis section. This is crucially important so readers are aware how the authors analysed each outcome measure. I would also include here information about power calculations and sample size estimates. If these were not conducted please provide solid justification.

Results

3.1 Demographics

Why do the authors not provide demographic information of the Italian and English controls in Table 1? This information is important to show exactly how matched samples were, especially for age, education, gender, VIQ, PIQ and FIQ.

Tense in the second sentence is in incorrect. Use did instead of do.

3.2 Cognitive performance

I found it very difficult to interpret the results in their current presentation, especially as stats aren’t given alongside the text. Could the authors consider presenting stats alongside text describing their significance and use the table to compliment the text?

There are no # displayed throughout Table 2. Were any of the stats in this table adjusted? If so, make it clear which stats were adjusted.

Figure 1 is small and difficult to follow. I would encourage making graphs larger, having both Italian and English data on one graph so the reader can make comparisons and panel each graph with a description of each in the figure caption.

Line 69-82. The latter part of this paragraph seems better suited to the discussion. Please only state findings in results and expand on them in the discussion.

Line 83-87. This is interesting and key to show that many PKU patients are mostly comparable to healthy counterparts.

3.3. Cognitive Outcomes in Relation to Metabolic Control

Line 89-98. Regardless of power it would be interesting to show whether correlations are similar for the English group. Both groups, were after all, the same size. How many patients were required for adequate power? Please capture this information. The authors can’t show the English correlations there must be stronger justification for this.

Table 4. Table caption mentions ‘Highlighted rows compare childhood Phe measures with current Phe’ but there are no rows highlighted.

Line 104-112. Did the authors adjust for the number of correlations conducted (n=144) as was performed with other outputs? Calculating numerous correlations increases the risk of a type I error, i.e., to erroneously conclude the presence of a significant correlation. To avoid this, the level of statistical significance of correlation coefficients should be adjusted. I would encourage the authors to do this and re-report the stats. If it has already been performed then this should be clear in the text.

General

The referencing style does not reflect that permitted by Nutrients. References must be numbered in order of appearance in the text

As per the abstract, many terms are in italics in places but this is not consistent throughout the manuscript. Please ensure consistency throughout.

It is clear the authors have double spaces to start new sentences, but there are areas throughout the manuscript where this rule is not followed. Please ensure consistency throughout.

Please fill out information concerning supplementary material (or remove sentences if no supplementary materials are submitted), author contributions, funding, acknowledgements and conflicts of interest.

Author Response

Comparing two nations, the present manuscript tested cognitive outcomes in Italian and English adults with PKU. This is an important area of study in the context of PKU as cognitive outcomes in relation to metabolic control remains to be fully understood, especially from a multi-country perspective. While it is difficult to standardise research methods in multi-country trials, the author’s should be commended for closely matching cohorts according to gender, age and education.

It was however very difficult to provide peer review as the submitted manuscript does not follow MDPI formatting guidelines. Line numbers were inconsistent; only starting at page 8 (section 3.2), citations were incorrect and revision of language is encouraged.

We have inserted Tables and figures in text and reorganized the order of references in text.  Please note thatNutrients now accepts free format submission, with references in any stile provided that they are consistent.  Line numbers are automatically provided, we will check that they start from the first page in the new submission.

While there is merit in the submission I feel it will be hard for readers to follow in its current state. By using the statement ‘recruitment stopped when enough participants were tested’ it seems recruitment was stopped, and findings submitted when it suited the authors intention. There is no mention of sample size calculations to facilitate whether the study was sufficiently powered. The manuscript could be condensed in placed and some key information is missing.

Power calculations showed that we needed 20 participants in the clinical group and 20 participants in the control group for a one tail effect size of .8 (consistent with what found in our previous studies) and a=.05, power (1-b error probability) =.80.  After PKU participants were contacted, one participant became unavailable and we were left with 19 participants still with acceptable power (=.78). 

Title

While the use of well-known abbreviations is permitted in scientific write up I would argue that the Phe is less known compared to PKU. I would therefore encourage writing phe in full and abbreviating phenylketonuria.  Phe has been written in full in the title.

Telephone for corresponding should start +44 (0) 121… This has been corrected

Abstract

Objective Objective is clear

Methods

‘From our original battery, we selected a smaller set of tasks that was still large and comprehensive enough to cover’. Please reword the first part of the sentence that introduces your methods. At present the text is rather informal.

This has been rephrased

Tasks were closely matched in English and Italian’. Do the authors mean language? If so be clear.

We make clear that when verbal stimuli/or responses were involved they were closely matched for psycholinguistic variables (e.g., word frequency, length)

Results

‘The Italian group had slightly worse metabolic control but showed levels of performance and patterns of impairment similar to the English group.’ I imagine the authors are referring to AwPKU here? Please ensure this is clear as it is difficult to know whether you mean the Italian and English group combined, AwPKU only or controls only

Thank you, this has been clarified.

As this is likely cited in the text the reference is not necessary remove.  Reference has been removed.

General comments on abstract

As per the instructions for authors I would encourage the headings are removed from the abstract. Headings have been removed

The terms English, Italian, matched and same are in italics in places but this is not consistent throughout the manuscript. Please ensure consistency throughout.  Italics has been removed

Write phenylalanine in full prior to abbreviating.  This has been done

Introduction

Paragraph 1.

Although the authors direct the readers to reviews regarding IQ and cognitive outcomes it would be good to offer a brief overview to provide context to readers of these findings.

We have provided more information.

While the authors provide details concerning average Phe levels in AwPKU most of the literature cited is 20+ years old. I would encourage citing more contemporary data that reflects the countries being studied in the manuscript. For example, Green at al recently showed average phe levels for UK patients which differed according to dietary adherence.

We have now eliminated the statement that groups of AwPKU studied in the literature have current Phe levels >1000 mmol, as part of our re-writing of the introduction.

Points 1-3 need to be re-formatted.

Paragraph 3

A space needs to be added between Phelevels.  This has been corrected

Paragraph 4

When talking about negative effects of PKU on cognition why are some words underlined? This is not needed.  Underlying has been removed

Overall, the introduction is long and difficult to follow. I would encourage condensing and focusing only what is relevant to the need for the submitted study. We have extensively rewritten the introduction making it more coherent and focused.

Methods

2.1 Participants

Present the time for diagnosis for both countries in days or hours for constancy.  Yes, this has been changed

Thank you for providing detail about how the cohorts were selected and matched.

Was this research facilitated by a power calculation to identify necessary cohort numbers? The statement. From the statement ‘Recruitment stopped when enough participants were tested’ it seems recruitment was stopped, and findings submitted when it suited the authors intention. 

We have re-written this section.  Power calculations showed that we needed 20 participants in the clinical group and 20 participants in the control group for a one tail effect size of .8 (consistent with what found in our previous studies), a=.05 and power (1-b error probability) =.80.  After PKU participants were contacted, one PKU participant became unavailable and we were left with 19 participants per group still giving our study acceptable power (=.78).

2.2. Metabolic measures

Although details are provided about metabolic measures, there is no mention of how blood samples were taken. Was this the same method across both cohorts? What is the intraclass correlations on the labs that quantified sample?

We added details: “For both the English and the Italian participants metabolic measures were taken regularly since birth and extensive records were available. Number of measures did not differ between countries (see Table 1).  Blood Phe monitoring was performed on dry blood spot collected after overnight fasting by High Perfrmace Liquid Chromatografy until 2010 and then via tandem mass spectroscopy.  The laboratories of both centres have adhered and contributed to International quality control systems.”   

 Typically, how many samples were averaged across the three age bands? If numbers are hugely different this may impact findings.  

Please note that number of metabolic observations is included in Table 1 and there are no differences between the two languages for any age band or across the lifetime.

2.3. Cognitive assessment

As IQ quantification differed across cohorts, have both IQ methods been validated and are the tools comparable? If so mention this and if not add this to limitations.  

Both of these IQ measures are validated.  The subsets of the WASI strongly correlate with the FSIQ measure from the WAIS and there is a strong correlation between overall IQ measured with the WAIS and the WASI (see also Schrimsher, O'Bryant & Sutker, 2008).  This has been specified in the revised text.

There is a lot of information in this section and while this is important for reader repeatability, could the authors consider condensing this section so it is easier to follow. The authors could consider supplementary material.

We understand the comment of the reviewer, but we feel that the description of the text (which is quite condensed) should be part of the main text since the nature of the tests is central for the aims of the paper. 

There is no statistical analysis section. This is crucially important so readers are aware how the authors analysed each outcome measure.

We have now included a data analysis section explaining our statistical analyses.  This should improve readability.

I would also include here information about power calculations and sample size estimates. If these were not conducted please provide solid justification.  

Information about power calculation are now included in the demographic section.

Results

3.1 Demographics

Why do the authors not provide demographic information of the Italian and English controls in Table 1? This information is important to show exactly how matched samples were, especially for age, education, gender, VIQ, PIQ and FIQ. 

We have now reformatted the tables.  Table 2 now provides results for both the Italian and English PKU groups and the relative controls (both demographics and cognitive performance).  Table 3 compares z scores for the Italian and English PKU groups

Tense in the second sentence is in incorrect. Use did instead of do.  The sentence has been corrected; however, the tense refers to what was shown in the table so here and in following sentences the present tense is used.

3.2 Cognitive performance

I found it very difficult to interpret the results in their current presentation, especially as stats aren’t given alongside the text. Could the authors consider presenting stats alongside text describing their significance and use the table to compliment the text.

We recognize that our presentation of results was not ideal and significance of results may have been difficult to take in.  To improve presentation, we have reformatted and added information to the tables.  Now Table 2 present each of the PKU group (Italian and English AwPKU), relative controls and statistical difference next to one another.  Hopefully this improves readability and makes easier to appreciate significant results, avoiding the need to duplicate results in text.

There are no # displayed throughout Table 2. Correction is displayed for Trail Making B-A

Were any of the stats in this table adjusted? If so, make it clear which stats were adjusted.  Statistics still significant after adjustment are preceded by the diacritic # as described in title of the table

Figure 1 is small and difficult to follow. I would encourage making graphs larger, having both Italian and English data on one graph so the reader can make comparisons and panel each graph with a description of each in the figure caption.  We have made the figure larger.  We have put the Italian and English groups on different panels but with the same scale, this is less confusing.

Line 69-82. The latter part of this paragraph seems better suited to the discussion. Please only state findings in results and expand on them in the discussion.  We have moved this paragraph to the general discussion.

Line 83-87. This is interesting and key to show that many PKU patients are mostly comparable to healthy counterparts.  Thank you, we agree.

3.3. Cognitive Outcomes in Relation to Metabolic Control

Line 104-112. Did the authors adjust for the number of correlations conducted (n=144) as was performed with other outputs? Calculating numerous correlations increases the risk of a type I error, i.e., to erroneously conclude the presence of a significant correlation. To avoid this, the level of statistical significance of correlation coefficients should be adjusted. I would encourage the authors to do this and re-report the stats. If it has already been performed then this should be clear in the text. 

Correcting for multiple correlations is important to avoid the possibility that an individual difference is significant by chance.  For example, this is important when the purpose is to establish whether an individual cognitive function is impaired in PKU.  In comparing AwPKU and controls, we have reported significance with and without correction; the same we have done in comparing z-scores of Italian and English AwPKU.  However, when we report correlations with Phe, we are not interested in the significance of an individual correlation (which will not be very reliable given the small N), but instead in demonstrating that there is a positive association between metabolic control and cognitive outcomes across metabolic measures and cognitive tasks.  For this analysis, nothing hinges on the significance of an individual correlation.  Instead, sensitivity of our tasks is demonstrated by the fact that the great majority of the correlations are positive (against a chance expectation of 50% positive and 50% negative) and the average correlation is significantly higher than 0. There is Plenty of power for these analyse

The power for c2with N=144 observation; effect size=.5; one tail is .9999.  The power for one-sample t-test with N=144 observation; effect size=.5;is also .9999.

We have explained statistical analyses in our newly added data analysis section.

Line 89-98. Regardless of power it would be interesting to show whether correlations are similar for the English group. Both groups, were after all, the same size. How many patients were required for adequate power? Please capture this information. The authors can’t show the English correlations there must be stronger justification for this. 

Correlations for a larger English sample with the same tests have been reported before (Romani et al., 2017).  Reporting correlation with a smaller less reliable sub-sample which may be different from those reported with a large sample seems misleading.  It will have the unwarranted consequence to drive the attention of the reader on the significance or lack of significance of individual correlations which proves unreliable with a larger sample.  We understand, however, the point of the reviewer that it may be of interest to compare correlations obtained from samples of Italian and English AwPKU of equal size.  In the revised manuscript, therefore, we also report for comparison the % of positive correlation, and the average correlation obtained with the English sub-sample.

Table 4. Table caption mentions ‘Highlighted rows compare childhood Phe measures with current Phe’ but there are no rows highlighted.  Rows have been highlighted by greying them, this has been clarified.

General

The referencing style does not reflect that permitted by Nutrients. References must be numbered in order of appearance in the text. In the instructions for authors it is stated Nutrients now accepts free format submission: references may be in any style, provided that you use the consistent formatting throughout. It is essential to include author(s) name(s), journal or book title, article or chapter title (where required), year of publication, volume and issue (where appropriate) and pagination. DOI numbers (Digital Object Identifier) are not mandatory but highly encouraged.

As per the abstract, many terms are in italics in places but this is not consistent throughout the manuscript. Please ensure consistency throughout.  Italics have been removed

It is clear the authors have double spaces to start new sentences, but there are areas throughout the manuscript where this rule is not followed. Please ensure consistency throughout.  We have used double spacing more consistently.

Please fill out information concerning supplementary material (or remove sentences if no supplementary materials are submitted), author contributions, funding, acknowledgements and conflicts of interest.   Information about author contributions, funding, acknowledgements and conflicts of interest is now included.

Reviewer 2 Report

This study investigates a number of cognitive tests in two cohorts of individuals with PKU vs controls. It focuses on adults with PKU. There is nothing particularly new about this study but it will be useful dataset for wider comparisons. The study design is appropriate and the conclusions are justified.

Author Response

We thank the reviewer for recognising the interest of our paper.

Reviewer 3 Report

I had the pleasure of reviewing your manuscript entitled “Cognitive outcomes and relationships with Phe in Phenylketonuria: A comparison between Italian and English adult samples”.  I read your paper with interest and think it is an important piece of work and a next step in harmonizing cognitive research in PKU, although I do believe an important step was missed prior to this and the English study  (i.e. no extensive systematic (based on existing literature) selection of the used cognitive test battery prior to this research, adding to more different cognitive tests to be used in this sample).

Having said that I believe research in PKU will benefit from more national and international multi-centre collaborations and agree with the need for common PKU testing batteries with an appropriate/comparable sensitivity across different countries/languages.

There are a few limitations with regards to the statistical analysis and interpretation (discussion) of your results I would like to point out:

  • Correlation does not mean causation. Especially given the small sample size (n=19) and large range of measures of metabolic control (e.g. concurrent Phe ranges from 65-1465), the likelihood that observed correlations are reliable is reduced. In order to help interpret these correlations (and also the reported differences in cognitive performance, see later bullet point) effect sizes should be reported. Effect sizes are crucial for the interpretation of observed differences between groups. Even though p-values indicate whether or not a significant difference exists, they provide no information about the magnitude of the difference.
  • It is widely accepted, however, that levels of Phe do affect cognitive outcomes, therefore it would have been more appropriate to perform ANCOVA(s) (instead of ANOVA(s)) with (measure of) metabolic control included as a covariate
  • In terms of observed differences in reaction times, even though significantly different, average differences are often small (e.g. 50msec). Again, in order to aid the interpretation of these findings, your results would benefit from the reporting and discussion of effect sizes. Also, in terms of day-to-day life, what kind of effect does such a small difference in effect sizes really mean?
  • Finally, there is no mention of any limitations of your research in the discussion (e.g. small sample size with wide variability (range) in concurrent phe and how this could affect (the interpretation of) your results).

Finally just a few small textual comments for consistently:

  • Second to last paragraph of introduction: “were ?? as close a possible” should be “were matched as closely as possible”. I feel like the word “participant” or “AwPKU” was missed several times in introduction too, suggest to proofread carefully
  • In table 1: fluctuation/variation is used interchangeably, please stick to 1 term for consistency; also remove “(SD)” behind fluctuation under lifetime metabolic measures.

Overall, I think this is an important piece of work, but the limitations with regards to the analysis and interpretation of results need addressing. Happy to review again after the paper has been updated.